# XLRS Rat with Rs1^-/Y^ Exon-1-Del Shows Failure of Early Postnatal Outer Retina Development

**DOI:** 10.3390/genes13111995

**Published:** 2022-10-31

**Authors:** Eun-Ah Ye, Yong Zeng, Serafina Thomas, Ning Sun, Zeljka Smit-McBride, Paul A. Sieving

**Affiliations:** 1Department of Human Anatomy and Cell Biology, University of California Davis, Davis, CA 95616, USA; 2National Eye Institute, National Institute of Health, Bethesda, MD 20892, USA; 3Department of Ophthalmology, University of California Davis, Davis, CA 95616, USA

**Keywords:** Rs1-KO rat, electroretinogram, optical coherence tomography, gene therapy, ectopic nuclei, retinal development, photoreceptor cells

## Abstract

We generated a Long Evans transgenic rat with targeted deletion of the whole Rs1 exon-1 and evaluated the pathological retinal phenotype of this *Rs1^-/Y^* rat model of X-linked retinoschisis (XLRS). The *Rs1*^−/Y^ rat exhibited very early onset and rapidly progressive photoreceptor degeneration. The outer limiting membrane (OLM) was disrupted and discontinuous by post-natal day (P15) and allowed photoreceptor nuclei to dislocate from the outer nuclear layers (ONL) into the sub-retinal side of the OLM. Dark-adapted electroretinogram (ERG) a-wave and b-wave amplitudes were considerably reduced to only 20–25% of WT by P17. Microglia and Müller glial showed cell marker activation by P7. Intravitreal application of AAV8-RS1 at P5–6 induced RS1 expression by P15 and rescued the inner nuclear layer (INL) and outer plexiform layer (OPL) cavity formation otherwise present at P15, and the outer-retinal structure was less disrupted. This *Rs1*^−/Y^ exon-1-del rat model displays substantially faster rod cell loss compared to the exon-1-del Rs1-KO mouse. Most unexpected was the rapid appearance of schisis cavities between P7 and P15, and then cavities rapidly disappeared by P21/P30. The rat model provides clues on the molecular and cellular mechanisms underlying XLRS pathology in this model and points to a substantial and early changes to normal retinal development.

## 1. Introduction

Human X-linked retinoschisis (XLRS) is an inherited retinal dystrophy with a prominent macular component that usually presents with reduced acuity in young boys by pre-school age. XLRS exhibits slow retinal degeneration over a lifetime, primarily in males, while heterozygous female carriers remain asymptomatic with only rare exception. Disease prevalence is 1 in 5000 to 25,000 worldwide. XLRS is one of the more common young-age onset genetic retinal dystrophies with the principal pathology including structural splitting involving the inner retina with the formation of “schisis cysts” and a characteristic electroretinogram (ERG) dark-adapted response with relative preservation of the photoreceptor a-wave but subnormal b-wave from post-photoreceptor bipolar cells [1]. The retinoschisin protein is encoded by the *RS1* gene and plays a key role in XLRS [2]. Numerous mutations in human *RS1* lead to dysfunctional RS1 protein and disrupt the structural integrity of retinal layers. RS1 is also required for proper synaptic signaling between photoreceptors and bipolar cells, which accounts for the sub-normal ERG b-wave relative to a-wave in XLRS [3,4,5]. These are key features to examine when recreating the human disease phenotype in a rodent model. 

Several *Rs1*^−/Y^ transgenic mouse models of XLRS [5,6,7] successfully mimic the structural and ERG functional deficits observed in human patients. These mice exhibit cystic schisis across the retina, as do many human XLRS males [1]. We previously developed an XLRS mouse model by inserting a neomycin cassette to replace Rs1 exon 1. This exon-1-del Rs1-KO mouse produces no Rs1 protein, and it shows fidelity to the human XLRS phenotype, with slow progression, reduced ERG b-wave amplitude relative to the a-wave, and retinal schisis cavities primarily in the outer plexiform layer (OPL) and inner nuclear layer (INL) [5]. 

The Rs1-KO mouse was essential for developing pre-clinical data for FDA regulatory oversight to initiate a human Phase 1/2a RS1 gene therapy trial [3,4,5]. Other Rs1-mutant mouse models have been generated by knockout or point knock-in mutations, and these showed a range of onset, progression, and severity [8]. Some showed early age retinal cavities with rapid degenerative changes, indicating phenotype variations between mutant models [9]. Consequently, questions remain unanswered regarding molecular, cellular, and pathophysiologic disease mechanisms in the XLRS retina. 

Transgenic rats have been employed to study ocular disease conditions [10,11,12,13]. The larger eye size of rat facilitates experimental manipulation more readily than the small mouse eye. Given the range of disease onset age and progression in the several XLRS mouse models, we also sought to evaluate the disease characteristics an Rs1-KO rat model with Rs1 exon 1 deletion comparable to our earlier exon-1-del Rs1-KO mouse. Here, we report key findings of this Rs1-KO rat model and identify several features in common or different from the Exon-1-del XLRS mouse model and human XLRS.

## 2. Materials and Methods

Transgenic and WT littermate Long-Evans male and female rats were used in this study. Transgenic *Rs1*-exon1-knockout (Rs1-KO) rats were generated in Long Evans rat using CRISPR/Cas9 to delete Exon 1. Animals were housed under normal 12–12 h lighting conditions in standard animal laboratory facilities maintain by the University of California, Davis. Ages and numbers of animals for each experiment are provided in the text or figure legends. The same animals were not tracked over time for either histology or ERG, as the histology is terminal, and ERGs were performed on separate age cohorts. 

The study was conducted according to the Guidelines of The Association for Research in Vision and Ophthalmology (ARVO) for the use of animals in Ophthalmic and Vision research and approved by the Institutional Animal Care and Use Committee (IACUC) of UC Davis (protocol code #21664 and date of approval: 24 March 2020).

### 2.1. Generation of Rs1h-Knockout Rats

The *Rs1*-exon1-knockout (Rs1-KO) was generated in Long Evans rat using CRISPR/Cas9 to delete Exon 1. Exon 1 was also the deletion site of our previous *Rs1h*-knockout mouse [5]. The Rs1-KO rat was generated with assistance from Horizon Discovery (Saint Louis, MO, USA). Two sgRNAs were targeted to delete ~769bp spanning exon 1 to partial intron 1 at genomic coordinates (794–1562 on Chromosome X_ NC 005120) that contain the murine orthologue of the human *Rs1* gene. Primers flanking the sgRNA sites were designed to test Nonhomologous End Joining (NHEJ) activity. Upstream primers were US-F (5′-TGTCGGAGAATTAGGGACCCA-3′) and US-R (5′-GCACGTACCTTCATAGCCAAA-3′). Downstream primers were DS-F (5′-AGCATAAGGCCAAGTACCACT-3′) and DS-R (5′-ACTAGGACTGCAGGTATGTATAGT-3′). The desired deletion between the two sgRNA sites was identified using primers US-F and DS-R, which gave a 500 bp PCR band for the KO allele, while primer pair US-F and US-R gave a 367 bp band identifying the WT allele. The founder rat was crossed to Long-Evans WT to generate F1 heterozygous female F1 animals that were bred to WT Long Evans males to generate F2 (*Rs1^−/Y^*) males. 

Rs1-KO rat tail genomic DNA (gDNA) was genotyped by PCR using primers US-F and DS-R to identify the targeted and WT allele. Long-range PCR was performed in a 50 μL reaction containing 200 ng of tail gDNA, 1 μM final concentration of each primer, and 1X Hot-Start GoTaq DNA polymerase (Promega). Conditions for amplification for both alleles were: 2 min at 95 °C for initial denaturation, followed by 30 cycles of 30 s at 95 °C, 30 s at 55 °C, 40 s at 68 °C, and 5 min at 68 °C for the final extension. Initially, PCR products were analyzed by agarose gel electrophoresis, while later, Transnetyx (https://www.transnetyx.com/ (26 August 2022)) was employed to do routine genotyping tests. 

### 2.2. Western Blotting

Retinas were harvested and lysed in pH 7.4 RIPA buffer (50 mM Tris-HCl, 1% NP-40, 0.25% sodium deoxycholate, 150 mM NaCl, and 1 mM EGTA supplemented with Halt protease inhibitor cocktail; Thermo Fisher Scientific, Cat#: 78429). Protein concentration was determined with the bicinchoninic acid (BCA) method using the Pierce BCA Protein Assay Kit (Thermo Fisher Scientific, Cat#: 23225). Retinal lysates (50 μg protein) were loaded in 4–12% Bis-Tris gels (Thermo Fisher Scientific, Cat#: NP0321BOX), followed by blotting onto PVF membranes by wet transfer. The membranes were blocked in blocking buffer (LI-COR Biosciences, Cat#:927–70001) and incubated overnight at 4 °C with primary antibodies diluted in 1X PBST. Primary antibodies were rabbit anti-Rs1 (1:1000) and mouse anti-β actin (1:3000). PDVF membranes were rinsed 3 times in 1X PBST, incubated for 1 h at room temperature with anti-rabbit HRP-conjugated secondary antibody (Cell signaling, Cat#: 7074), and scanned on a Sapphire biomolecular imager (Azure Biosystems).

### 2.3. Optical Coherence Tomography (OCT) Imaging

OCT imaging was used to monitor retinal structural changes and cavity formation using an Envisu R2200 SD-OCT ophthalmic imaging system (Bioptigen, Durham, NC, USA). Rats were anesthetized with an intraperitoneal injection of ketamine (80 mg/kg) and xylazine (8 mg/kg). Pupils were dilated with topical 0.5% tropicamide and 0.5% phenylephrine HCl. Artificial tears (Alcon Laboratories, Inc., Fort Worth, TX, USA) were used throughout the procedure to maintain corneal hydration and clarity. Radial volume scans consisting of 10 B-scans (1000 A-scans per B-scan) were collected with an average of five frames each. Rectangular volume scans with 100 B-scans were collected across a 2.6 × 2.6-mm area centered on the optic nerve (ON) head. Two linear B-scans, an average of 10 frames each, were obtained from nasal to temporal pole through the ON head.

### 2.4. Electroretinogram (ERG)

Full-field ERGs were performed on WT and Rs1-KO rats after animals were dark-adapted overnight for 15 to 18 h. Preparations for ERG were made under dim red light. Rats were anesthetized with intraperitoneal ketamine (80 mg/kg) and xylazine (8 mg/kg), and the pupils were dilated with topical 0.5% tropicamide and 0.5% phenylephrine HCl. A gold wire loop electrode was placed on the cornea, and responses were referenced to an electrode underneath the tongue and a ground electrode in the tail. Body temperature was maintained at 37 °C during the recording session. Both eyes were recorded simultaneously. After finishing the recordings, lubricant eye gel was applied to the corneal surface to prevent drying, and rats were maintained on the heating pad until they were fully recovered. Scotopic responses were elicited to single flashes from −5.3 to +0.7 log cd·s/m^2^ in 0.5-log steps. Responses were filtered using a 1 to 500 Hz bandpass and a 60 Hz line-frequency filter. Photopic light-adapted ERG responses were recorded against a rod-saturating background (20 cd/m^2^), and flash intensities were −0.3, 0, 0.7, 1, 1.7, 2 log cd.s/m Recordings were made using an Espion E2 Electrophysiology System with a ColorDome Ganzfeld stimulus (Diagnosys LLC, Lowell, MA, USA). 

### 2.5. Intravitreal Injection of AAV8-RS1 Vector

scAAV8-hRS/IRBP-hRS vector was administered by intravitreal injection to pups at P5 or P6. This vector is comparable to the clinical vector in a human XLRS trial [14]. Pups were anesthetized either by 20–30 µL volume via intraperitoneal injections of ketamine/xylazine mixture containing ketamine hydrochloride (30–40 mg/kg) and xylazine (3–4 mg/kg) for 10–15 min or by isoflurane inhalation (1.5% isoflurane in 100% oxygen) for 3 to 5 min. One drop of 0.5% tetracaine topical anesthetic was applied, and pupils were dilated using a drop each of 1% tropicamide and 2.5% phenylephrine. One drop of Proparacaine hydrochloride 0.5% was applied as topical ocular anesthesia. A small scleral nick was made at the nasal-temporal ora serrata below the iris using a 30-G insulin syringe needle to gain access to the posterior cavity (vitreous chamber) of the eye. Sterile 10 µL Nanofil syringes with the 35 G beveled-tip needle on a sterile 10 µL Nanofil syringe (World Precision Instruments, Inc., Sarasota, FL, USA) was inserted into the vitreous to deliver 2e10 (2 × 10^10^) vector genomes per eye (vg/eye) in 1.5 μL fluid. The contralateral eye served as control. Triple antibiotic ophthalmic ointment (neomycin, polymyxin B, and bacitracin) was applied to the eye at the conclusion. Animals were kept on a warming pad until their recovery [15]

### 2.6. Histology and Immunohistochemistry

Rats were sacrificed by carbon dioxide asphyxiation. For paraffin sections eyes were oriented, enucleated, and fixed with 97% methanol/3% glacial acetic acid for up to 5 days prior to embedding in paraffin [15]. Sagittal sections 5 μm thick were cut through the eye and stained with hematoxylin and eosin (H&E). Retinal images were collected using a Nikon Eclipse e800 microscope with a DS-Ri1 digital camera (Nikon, Tokyo, Japan). For cryosections, eyes were oriented and hemisected. The eye was enucleated and fixed overnight in 4% paraformaldehyde (PFA) and then processed for cryosection. Sagittal cryosections were cut at 10 μm. Retinal cryosections for immunohistochemistry were washed in 1x PBS and preincubated with 4% normal goat serum and 0.5% Triton X-100 in 1x PBS, at room temperature (RT) for 1 h. Paraffin sections for immunohistochemistry were rehydrated prior to blocking, then washed with phosphate buffer 0.1% Tween 20 (1xPBST) and preincubated with (5% normal goat serum, 10% normal donkey serum, and 0.1% Tween 20 in 1xPBS) at RT for 2 h. Primary antibodies with appropriate dilution using blocking buffer containing 1xPBS were added (Table 1). 

For immunohistochemistry, retinal sections were incubated with primary antibodies overnight at 4 °C for cryosections and RT for paraffin sections. Retinal cryosections were washed with 1x PBST 3 times for 40 min each and paraffin sections were washed with 1x PBST 3 times for 15 min each. The fluorescent secondary antibodies (Alexa Fluor 488 donkey anti-mouse, Alexa Flour 488 goat anti-mouse, Alexa Fluor 488 donkey anti-goat, Alexa Fluor 647 donkey anti-guinea pig, Alexa Fluor 647 donkey anti-mouse, Alexa Fluor 555 donkey anti-rabbit, and Alexa Flour 568 goat anti-rabbit; Jackson ImmunoResearch Laboratories) at 1:1000 in PBST were added to retinal sections and incubated for 1.5 h. Tissue sections were washed in PBST 3 times for 15 min, coverslipped with DAPI Fluoromount-G® (SouthernBiotech, Cat#: 0100-20) mounting media and imaged on a Confocal Laser Scanning Module LSM 510 Microscope System (Carl Zeiss Microscopy, Jena, Germany) and the Olympus Fluoview FV3000 Confocal Laser Scanning Microscope (Olympus, Tokyo, Japan). Representative images were obtained from retinas from at least 3 animals every KOs and Heterozygotes for each age and at least 2 animals for every control age.

### 2.7. Statistical Analysis

Data were analyzed using Prism 5 (GraphPad Software, La Jolla, CA, USA) and presented as mean ± SEM. Two-tailed Student’s *t*-test was used to calculate group differences with Welch *t*-test. *p* values less than 0.05 were considered significant.

## 3. Results

### 3.1. Early Post-Natal Pathology of Rs1^-/Y^ Rat

The *Rs1*-exon-1-KO rat XLRS model was generated in Long Evans rat using CRISPR/Cas9 technology (Figure 1), with the upstream (US) and downstream (DS) single-guide RNA (sgRNA) surrounding exon-1. The *Rs1^-/Y^* rat model was designed to be genetically parallel to the exon-1-del of our previous Rs1-knockout mouse [5].

Retinal histology of WT and *Rs1^-/Y^* appear similar at post-natal day 7 (P7), with orderly lamination of the inner and outer retina, and initial development of photoreceptor inner- and outer-segments (IS/OS) as expected (Figure 2A). However, by P15 the retinal lamination and outer nuclear layer (ONL) cellularity changed dramatically, with very large INL cavities and ectopic photoreceptor nuclei displaced into the IS/OS region (*arrowheads,*
Figure 2A). They were displaced across the outer limiting membrane (OLM), which was no longer continuous (Figure 2A).

By P30, only two weeks later, the INL schisis had resolved, and only a few small cavities remained. Closure of INL schisis was mostly complete even by P21 (not shown). Ectopic photoreceptor nuclei touched the retinal pigment epithelium (RPE) and only approximately half or less of the photoreceptors remained in the ONL. Cellularity of the INL lamina at P30 remained similar to WT, but intrusion of cells from both the INL and ONL made the OPL lamina irregular. By 3.5 months, the displaced photoreceptor nuclei and cells had mostly disappeared, making the ONL lamination appear more regular again but with major loss of photoreceptor soma. ONL degeneration continued out to 10 months when only approximately 10% of the normal number of nuclei remained. 

OCT imaging confirmed these changes in vivo albeit at lower resolution (Figure 2B). The WT retina showed clearly separated laminar structure of the inner retina, OPL, ONL, OLM, and EZ [3,4,5]. *Rs1^-/Y^* mouse OCT imaging was difficult at P15 because the giant schisis cavities within the INL obscured clear imaging of the ONL and other structures. However, by scanning multiple retinas (including that shown in Figure 2B), the layers in the retina were seen to have merged or were discontinuous. These OCT changes remained evident as the retinal degeneration progressed with age. By P21 and P30 no schisis cavities were evident, and ONL thickness of the *Rs1^-/Y^* decreased to nearly half of the WT. ONL thickness was further depleted by 3 months, IS/OS were shortened, and the OPL was disordered. However, the structure of the innermost retina appeared spared. The OCT and retinal histology concurred that the exon-1-del *Rs1^-/Y^* rat XLRS model exhibits very early, postnatal outer retinal disorganization which progresses rapidly.

### 3.2. Rs1^-/Y^ Rat Retinal Function by ERG 

Dark-adapted ERG recordings show major functional impairment as early as P17, with both the a- and b-waves reduced to only 20% of WT amplitudes (N = 3 animals each) (Figure 3A,B). Similar reductions were observed at 1 month when a- and b-wave amplitudes measured 18% of WT (* *p* < 0.05, N = 5) (Figure 3C,D). These small ERG responses decreased further by 6.5 months, with a- and b-waves only 17% and 12% of WT, respectively (* *p* < 0.05, N = 3) (Figure 3G,H). The major, early a-wave loss indicates substantial photoreceptor dysfunction even by young postnatal age. The a- and b-waves appeared reduced proportionately, contrary to some other photoreceptor degenerations, e.g., P23H rhodopsin transgenic rat in which photoreceptor loss initially causes a-wave reduction with relatively b-wave sparing [13]. By comparison the *Rs1^-/Y^* rat has b-wave loss equal or greater than a-wave loss, possibly indicating dysfunction beyond the photoreceptors alone, such as OPL synaptic dysfunction identified for XLRS mouse models, including the exon-1-deletion *Rs1^-/Y^* mouse. However, drawing firm conclusions with these small responses warrants caution.

### 3.3. Disruption of OLM Integrity

As ectopic photoreceptor nuclei were seen in the IS/OS by P15, we examined OLM integrity in WT and Rs1-KO rats using antibody against b-catenin which localized to ONL adherens junctions (AJs). The OLM of WT rat formed a continuous line with β-catenin (green) labeling at P7, P15, and P30, and the photoreceptor nuclei remained within the ONL layer (Figure 4). The OLM of *Rs1^-/Y^* retina appears intact at P7, but by P15 it is fragmented, and photoreceptor nuclei are displaced into the IS/OS individually or in small clusters adjacent the OLM breaks. OLM disruption increased by P30. 

### 3.4. Rs1-KO Photoreceptor Development 

Photoreceptors IS/OS structures and bipolar synapses were imaged by co-localization with anti-opsin (OPN2), Rs1 and PSD95 antibodies. Opsin expression in WT rat begins at P5 [16,17] with retinal development completed by P21 [18]. Early development of *Rs1^-/Y^* photoreceptors at P7 progressed normally with opsin expression (Figure 5). WT photoreceptors and bipolar cells show Rs1 expression by P7, while, of course, the *Rs1^-/Y^* retina has no Rs1 expression at that age. PSD95 is minimally present at the presynaptic photoreceptor terminals at P7 in the WT and *Rs1^-/Y^* retinas (red arrowheads), but by P30, WT had strong PSD95 labeling of photoreceptor axon terminals (yellow arrowheads), along with co-labeling of Rs1 and labeling of the bipolar post-synaptic dendritic elements, all as expected. PSD95 labeling of the *Rs1^-/Y^* retina is sparse, but this is present at P30 (white arrowhead).

### 3.5. Microglial and Müller Glial Activation in Rs1^-/Y^ Retina 

Microglial activation is a feature of retinal degeneration [19]. In *Rs1^-/Y^* rat, microglia/macrophage activation is seen early in postnatal retinal development (Figure 6A, Iba1 labeling, red). In the WT retina, the microglia and macrophages are located primarily in the proximal retina ganglion cell layer (GCL), inner plexiform layer (IPL), and INL, while the *Rs1^-/Y^* rat retina showed a slight increase of Iba1 immunoreactive cells as early as P7. Microglial activation increased by P15 and P30 with a number of ameboid morphology that have migrated into the ONL and IS/OS region and may be phagocytosing the ectopic photoreceptors in the deep retina. Müller glia activation was also evident in *Rs1^-/Y^* retina, with increased GFAP expression (green) at P15 and P30. Microglial changes at P7 appear to precede Müller glial activation in *Rs1^-/Y^* retina. By P30, SOX9 labeling (red) shows substantial Müller cell loss in the *Rs1^-/Y^* retina (Figure 6B). 

### 3.6. Female Rs1 Transgenic Rats

Retinal histology of *Rs1^-/+^*and *Rs1^-/-^* female rats were compared with *Rs1^-/Y^* males at P30 and 10 months (Figure 7). Heterozygous *Rs1^-/+^* females showed no alterations of normal retinal laminar structure compared with WT *Rs1^+/+^* at either age. However, homozygous knockout *Rs1^-/-^* females had the abnormalities previously seen in *Rs1^-/Y^* mutant males: by P30 some photoreceptor nuclei were ectopically displaced into the IS/OS layer, and only a few small residual schisis cavities remained. Only approximately half photoreceptor nuclei remained compared to the normal WT *Rs1^+/+^* retina. Retinal thickness decreased further by 10 months with only 1 to 2 layers of ONL remaining. The effect of age on layer thinning had been seen in the male *Rs1^-/Y^* from ages 6–10 months (Figure 2A). Overall, the histologic retinal changes in homozygous *Rs1^-/-^* female rats mimicked what was observed in the *Rs1^-/Y^* male for these points, and as in the *Rs1^-/Y^* males, the cavities had collapsed in the *Rs1^-/Y^* male by P30.

### 3.7. Rescue of Rs1^-/Y^ Retina by AAV8-RS1

Rescue of the retina of *Rs1^-/Y^* rat by RS1 gene delivery was tested by intravitreal application of scAAV8-hRS/IRBP-hRS vector, 2 × 10^10^ viral genome (vg)/eye, and the untreated contralateral eyes were controls (Figure 8). Vector application was performed at P5/P6 before the prominent phenotype changes had occurred. OCT imaging at P15 and P30 showed a reduction of retinal cavities after treatment, and the retinal lamellar structure was considerably better preserved (Figure 8A). Compared with contralateral control eyes, the AAV8-RS1 treated retinas showed a reduction or absence of cavities by P15 and P30 (n = 7). Treated eyes showed better laminar structure, including in one case restoration of the four normal outer reflective retinal bands (ORRB) on OCT imaging (*box*). Retinal histology at P30 after application of AAV8-RS1 vector showed substantial structural rescue in some cases (Figure 8B, arrowhead) with RS1 expression in photoreceptor IS comparable to WT (Figure 5, WT at P30). Treated retinas displayed an intact OLM (b-catenin) at the IS-OLM junction. Rhodopsin expression in rod OS after treatment was preserved. Microglia (Iba1) were absent from the ONL as expected for WT. Unlike the untreated *Rs1^-/Y^* retina, Müller glial cell numbers (SOX9) after RS1 expression remained near normal (compared with WT, e.g., see Figure 6B, WT at P30), and Müller glial had a lower activation state (GFAP) (Figure 6B and Figure 8B). These results provide further evidence of the therapeutic potency of the AAV8-RS1 vector in *Rs1^-/Y^* retina, particularly when administered at a young age.

## 4. Discussion

This *Rs1^−/Y^* exon-1-del rat model of XLRS shows retinal abnormalities by early postnatal age with rapidly progressive pathology. Nearly half of the photoreceptors are lost from the ONL by P30. Photoreceptor loss then slows considerably, with an additional further loss out to 10 months of age. One mechanism of ONL cell loss involves escape of photoreceptor nuclei into the IS/OS zone between the OLM and the RPE. Consonant with the early age structural changes, both the a-wave and b-wave ERG amplitudes are 80% reduced by P17. Application of therapeutic AAV8-RS1 at early age of P5–P6 provided some rescue from structural disorganization of the retinal layers. 

### 4.1. Comparing Two Rs1-KO Rat Models of XLRS

We previously created an XLRS rat model using CRISPR/Cas technology to disrupt Rs1 exon-3. This gave an early onset and rapidly progressive XLRS phenotype, with development of major INL schisis cavities by P15, followed by rapid collapse of cavities by P21- P30 [20]. Although no off-target effects were identified, we were concerned for some unidentified off-target effect, as the XLRS phenotype diverged from our expectations based on existing XLRS mouse models. Hence, we created this second *Rs1^-/Y^* rat by targeting exon-1 in search of a slower and more modulated phenotype, homologous to the slowly progressing Rs1-KO exon-1-del mouse model we had created previously [5]. However, the phenotype of the Rs1 exon-1-del rat mirrored the Rs1 exon-3 rat, as both differ from the majority of murine XLRS models: both rat models have early onset of extensive retinal structural pathology, rapid ONL cell loss, early development of INL large schisis cavities by P15, and then rapid closure of the schisis cysts by P30, and both XLRS rat models have an overlapping ERG phenotype with major dark-adapted response reduction very early by P17–P21. Both exon-1-del and exon-3 rat models were partially rescued by early restoration of RS1 using the therapeutic AAV8-RS1 vector. Although two examples of XLRS rat models do not exclude other possibilities, the very rapid XLRS phenotype in rat may represent a species difference from mouse. These rat models may have a role in understanding the biology of Rs1 and in developing human gene therapy for this condition. Further investigations are needed to understand how Rs1 protein participates in development at early post-natal ages.

### 4.2. Comparison of RS1-Exon-1 Del Rat with Human XLRS

Human XLRS disease historically was considered a “stationary” retinal dystrophy [21] rather than progressive. Many young human XLRS subjects have normal or near-normal ERG a-wave amplitudes, indicating overall normal photoreceptor numbers and outer/inner segment length [22]. However, the ERG a-wave of some human XLRS subjects can be reduced by early age [22], and OCT imaging of the outer retina in younger XLRS boys can show reduced ONL width [23], as we found in this new Rs1 exon-1-del rat. Visual acuity loss correlated with disorganization of the photoreceptor EZ band on OCT imaging in some younger boys, consistent with vision loss from outer retinal structural abnormalities separate from schisis [23]. Human XLRS studies have shown a correlation of visual acuity loss with shorter IS/OS length on OCT imaging, implicating structural photoreceptor changes in limiting vision [24]. Hence, the XLRS rat model reflects this feature of human XLRS disease as having a developmental component. 

### 4.3. Comparison with Exon-1 Rs1^−/Y^ Mouse

This XLRS rat and our original XLRS mouse were both created by deletion of exon-1. CRISPR/Cas9 technology was used for the rat, while in mouse, a neomycin cassette was inserted to replace exon-1 [5]; both models have an absence of Rs1 protein. 

The exon-1 Rs1-del mouse was one of the slower progressing XLRS phenotypes among the 7 mouse models created thus far [8]. Hence, it was surprising to find early severity and rapid progression for the exon-1-del *Rs1^−/Y^* rat.

As Table 2 indicates, both structural and ERG functional parameters of the Rs1 exon-1-del XLRS rat diverged markedly from the homologous XLRS mouse, with earlier and more severe phenotype even during the completion of retinal development up to P21. There are considerable differences in the acute dynamics of the structural schisis, with maximal, large OPL/INL schisis cavities appearing by P15 in rat but usually later in mouse. Cavity size of the Rs1 exon-1-del mouse reached maximum by 4 months and then decreased gradually out to 8 months [25]. In marked contrast with the homologous mouse, the Rs1 exon-1-del rat shows no cavities at P7, huge ONL cavities at P15, and coalescence by P21–P30. These acute structural changes are considerably accelerated in Rs1 exon-1 del rat compared with the homologous XLRS mouse.

Some of the XLRS mouse models also show rapid onset, e.g., three models (point mutant substitutions for C59S and for R141C, and a knockout from lacZ reporter gene substituted for exons 1–3) created by Liu et al. which showed major schisis cavity formation by P15 [9]. The schisis consolidated and cavities were reduced but still present out to 9–12 weeks and beyond. 

### 4.4. Functional ERG Loss Occurs Early

XLRS characteristically shows a disproportionately reduction of the dark-adapted ERG b-wave to the a-wave which is frequently preserved in human XLRS [21]. Some degree of dark-adapted a-wave reduction is common for mouse models, but nonetheless the b-wave loss is disproportionally greater, as in human XLRS [8]. However, few of the *Rs1^-/Y^* mouse models have such severe ERG loss by very early age as both XLRS rat models. XLRS rat has extensive early age a-wave loss, beyond that of mouse models and beyond all but the exceptional case in human XLRS. The Rs1 exon-1-del mouse had 33% a-wave loss but 50% b-wave loss at 1 month age, giving an “electronegative” configuration [25], and other Rs1-mutant mice showed electronegative ERG responses (i.e., b-wave loss > a-wave), from P15 to 28 weeks [9]. In contrast, the Rs1 exon-1-del rat has a massive, 80% reduction of both a- and b-waves by P17 and then decreased further to only 12% of WT over the following 6 months. The loss of proper IS/OS orientation of the displaced photoreceptors in the XLRS rat would reduce their contribution to the ERG a-wave, along with reductions of photoreceptor IS/OS length and ONL thickness, and the ERG loss appears to reflect the early retinal structural disorganization in the *Rs1^-/Y^* rat. 

### 4.5. Early OLM Disruption and Photoreceptor Aberrant Migration in Rs1 Exon-1-Del Rat Retina

Photoreceptor outer segments normally begin to develop at P4/P5 in rat [16], and this occurs in the Rs1-KO rat by the expected age, leading to normal retinal layer appearance by P7. However, in the XLRS rat, photoreceptor nuclei show anomalous migration after P7 and begin to move through the OLM and into the IS/OS region, and by P15 many ectopic ONL cells are in the IS/OS. 

The OLM consists of a group of AJs proteins which integrate with apical processes of Müller glial cells and anchor the OLM by attaching to the apical base of photoreceptor inner segments [26]. Imaging of OLM AJs between Müller glia and photoreceptor inner segments using an anti-β-catenin antibody [27] shows OLM disruption by P15. The OLM is discontinuous and fails to maintain the normal placement of photoreceptors in the ONL, as prominent, abnormal movement of photoreceptor cells occurs into the IS/OS zone. By P30, large clusters of displaced ONL cells occupy the IS/OS, and the ONL has lost nearly half of the photoreceptors. We are unclear of the mechanisms that contribute to the loss of cells in the ONL. For comparison, human XLRS can show a broad range of ERGs even in younger age, from nearly normal a-waves to considerably reduced responses [22], consistent with a range of photoreceptor numbers, from normal to reduced. 

Displacement of photoreceptor nuclei through the OLM have been observed previously in the exon-1-del mouse retina during postnatal development and into adulthood [5,25]. It also occurs in our Rs1 exon-3 rat model [20]. 

The mechanism underlying the mislocalization of the photoreceptors is not known. At least two mechanisms are possibly involved: one, the OLM acts as a barrier, and without Rs1 the barrier is disrupted allowing photoreceptors to move; or two that without Rs1 protein the photoreceptors migrate on their own and disrupt the OLM integrity. We venture that the latter possibility is involved, as intrinsic movement of photoreceptors is known in the normal developing retina. In rat, photoreceptors have intrinsic capability to migrate at P9 to P15 [28]. In mouse, between P11 and P15, cone photoreceptors enter a normal developmental migratory phase and move towards and reside adjacent to the OLM [29]. Timing of these events coincides approximately with aberrant movement of photoreceptors in the *Rs1^-/Y^* rat. In such case, the photoreceptors have intrinsic ability to migrate, and in XLRS rat, apparently the normal OLM is a limiting barrier which fails without Rs1 protein. A plausible place for failure would be at the IS apical region where AJs form attachment to Müller cells. Rs1 is copiously present on the outer membrane leaflet of the IS where it could interact with the AJs and Müller cells. 

The Rs1-KO rat showed microglial activation at P7. Previous RNA-seq study of Rs1-KO mouse found dysregulation of immune response genes and a proinflammatory state driven by microglia [30]. Microglial activation can induce Müller glia to be reactive. Interactions between microglia and Müller cells are known to be bi-directional and involved in overall pathological responses in the retina, as the interaction amplifies the glial activation and helps mobilize microglia in the retina [31]. 

The Rs1-KO mouse showed degeneration mostly limited to the outer retina as compared to inner retina, Rs1 was expressed by photoreceptors and bipolar cells. Synapses in the IPL are formed earlier than OPL synapses in the mouse retina. Postnatal developmental effects of Rs1, which may exert after P7, might result in outer retina to be more vulnerable to the pathology of Rs1-KO.

### 4.6. Rescue by RS1 Gene Delivery

XLRS pathology can be abridged or reduced by RS1 delivery using a therapeutic AAV8-RS1 vector at very early age before total breakdown of the retinal ONL lamellar structure. This provides a clue to identifying an optimal therapeutic window by which gene therapy can exert a substantial rescue effect on the Rs1-KO retina. In this case, the gene delivery was prior to prominent phenotype changes and achieved noticeable rescue from photoreceptor loss in the Rs1-KO rat. Vector delivery by intravitreal application must cross the ILM to enter the retina. We do not know whether the ILM of the XLRS rat reaches adult characteristics by P7, but the vector reached both the bipolar cells and photoreceptors, as they show Rs1 expression in the Rs1-KO retina. The outcome suggests that RS1 gene delivery suppressed the pathological glial responses to prevent or limit the progression of retinal degeneration. This rescue is comparable to the Rs1-KO mouse, for which AAV(2/2)-CMV-Rs1 delivered at P14 gave a long-lasting effect on retinal structure and function that persisted as late as 14 months [25]. 

In summary, XLRS pathology in *Rs1^-/Y^* rat resembles early and severe XLRS human disease. Several points can be extracted from having two examples of *Rs1^-/Y^* rat models:This exon-1-del *Rs1^-/Y^* rat shows very early, developmental retinal structural and functional pathology that mirrors the phenotype of the previous exon-3 *Rs1^-/Y^* rat. Neither of these rat models have expression of intact Rs1 protein.The phenotype of this exon-1-del *Rs1^-/Y^* rat is markedly earlier and faster than the exon-1-del *Rs1^-/Y^* mouse.Both exon-1-del and exon-3 *Rs1^-/Y^* rat XLRS models exhibit pathology considerably earlier and more progressive on average than most of the murine models. This may represent a species difference in rat versus mouse.The XLRS pathology of both Rs1 exon-1-del and exon-3-XLRS rat models can be abrogated or slowed by AAV-RS1 delivery of exogenous RS1 expression.These two *Rs1^-/Y^* XLRS rat models provide an opportunity to investigate retinal developmental pathology from lack of Rs1 protein expression in the retina.

## Figures and Tables

**Figure 1 genes-13-01995-f001:**
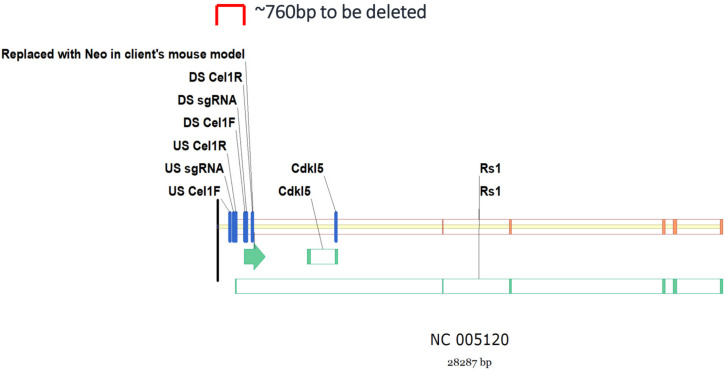
Generation of *Rs1^-/Y^* rat XLRS model. Rs1-knockout (Rs1-KO) rat was generated by CRISPR/Cas9 deletion of 769 bp (genomic coordinates 794-156 p on Chromosome X_ NC 005120) that contains exon 1 of the murine *Rs1* orthologue.

**Figure 2 genes-13-01995-f002:**
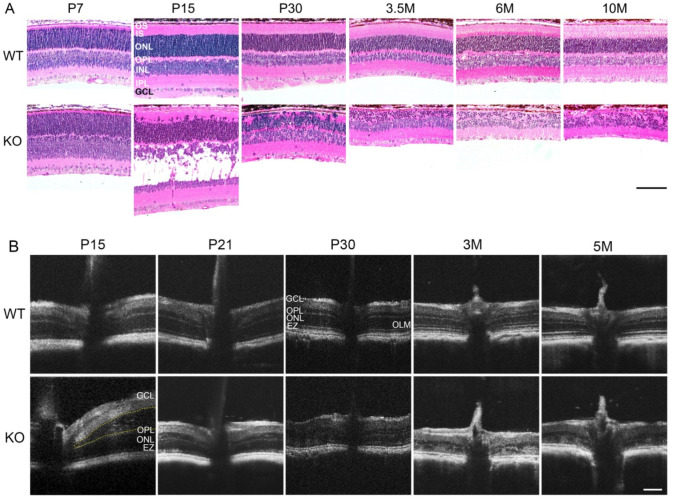
Retinal structural changes of *Rs1^-/Y^* rat. (**A**) Histology of WT and *Rs1^-/Y^* (KO) retinas. Retinas at post-natal day 7 (P7) showed essentially no differences in lamination at this magnification. At P15, large schisis cavities formed in the inner nuclear layer (INL) of the KO retina, and a few photoreceptor nuclei were dislocated into the photoreceptor inner segment (IS) zone. A greater number of the outer nuclear layer (ONL) cells (arrowheads) were dislocated into the IS/outer segment (OS) zones. At P30, only approx. 40% of the ONL cells remained, and the outer plexiform layer (OPL) was disrupted. From 3.5 months on, the thickness of the ONL was reduced, and only approx. 10% of the ONL remained at 10 months. The ONL lamination appeared more orderly as the displaced cells had mostly been eliminated. Scale bar = 100 μm. (**B**) Optical coherence tomography (OCT) imaging of WT and KO retinas. An extensive retinal cavity was observed in the INL starting at the optic nerve of KO rats at P15, but cavities were resolved by P21. KO retinas showed progressive degeneration in the outer retina with aging. ONL thickness of KO retina was reduced by approx. 60% at P30 compared to WT, and the ONL, OPL, outer limiting membrane (OLM) and ellipsoid zone (EZ) were disrupted. More severe degeneration was seen at 3 (3 M) and 5 months (5 M), and the retinal lamination was hardly evident compared to the WT. Scale bar = 200 μm.

**Figure 3 genes-13-01995-f003:**
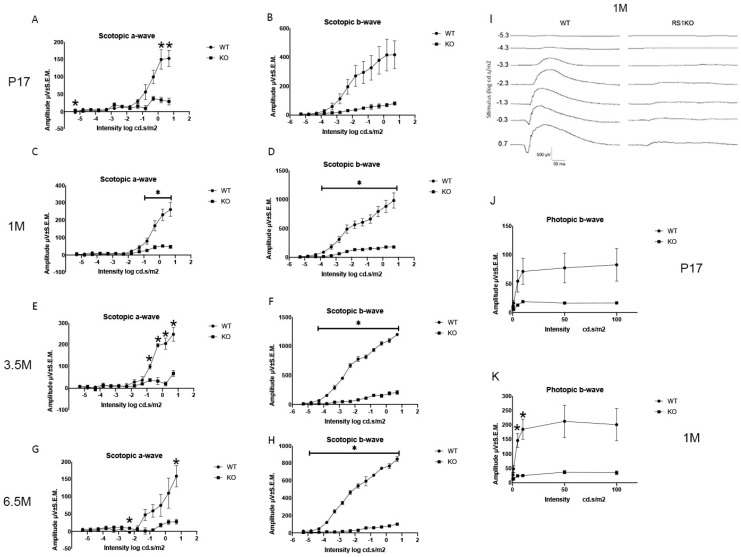
The functional analysis of *Rs1^-/Y^* rat retina by ERG. Dark-adapted rod scotopic and light-adapted cone photopic electroretinogram (ERG) responses of WT and Rs1-KO rats. All *Rs1^-/Y^* scotopic responses showed major reduction at all ages tested (* *p* < 0.05 throughout). (**A**,**B**). At P17, both the a- and b-waves of the *Rs1^-/Y^* were about 20% of WT amplitude (N = 3), and (**C**,**D**) at 1 month, amplitudes were 18% of WT (N = 5; * *p* < 0.05), and (**E**,**F**) these changed persisted at 3.5 month. (**G**,**H**) These reductions were maintained to age 6.5 months with little further reduction as a-, and b-waves remained at 17% and 12% of WT, respectively (N = 3; * *p* < 0.05). Note that the a-wave at 6.5 months was 17–18% of WT amplitude, while the post-synaptic b-wave was reduced further to 12% of WT, giving a (b-wave/a-wave) ratio of less than one. (**I**). Representative scotopic ERG waveforms at 1 month. (**J**,**K**) Photopic b-wave responses are reduced to 20% of WT at P17 and 17% at 1 M (N = 3).

**Figure 4 genes-13-01995-f004:**
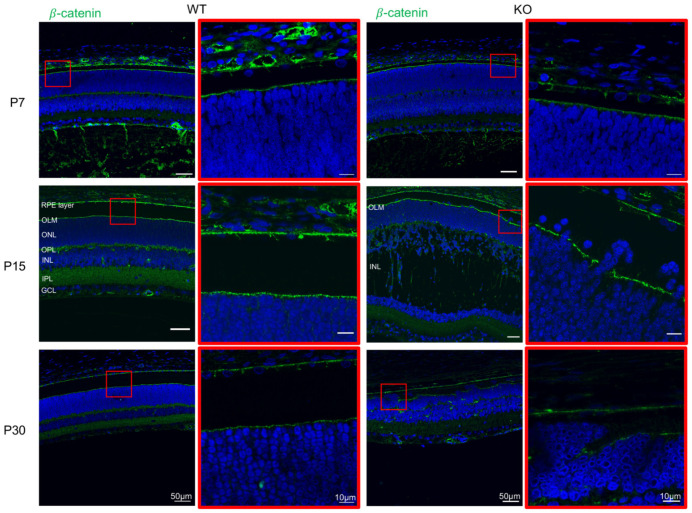
Disruption of outer limiting membrane (OLM) integrity in *Rs1^-/Y^* rat retina. Immunostaining of WT and *Rs1^-/Y^* (KO) rat retinas with β-catenin (green, OLM marker) at P7, P15 and P30. The images found on the right columns of WT and KO are a magnified portion of the image section encased in the red box in the left column. WT retinas showed a continuous outer limiting membrane (OLM) at all time points tested. In the KO retinas, the OLM also appeared continuous at P7. However, the initial presence of OLM breakdown was found at P15, and dislocated photoreceptors were found above the OLM at P15 and P30 in clusters of nuclei in the KO retinas.

**Figure 5 genes-13-01995-f005:**
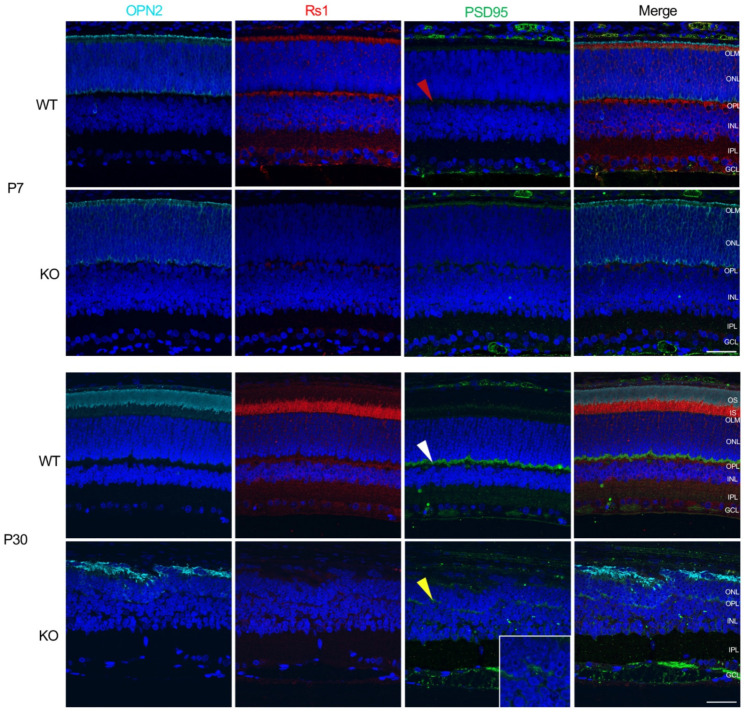
Photoreceptors and synapse immunolabeling in *Rs1^-/Y^* rat retina. Immunolabeling of WT and KO retinas for rhodopsin (OPN2), retinoschisin (Rs1), and post-synaptic density marker (PSD95) at P7 and P30 time points. Antibody labeling of OPN2 (cyan) in the outer segments of rod photoreceptor, Rs1 (red) in the inner retinal cells, and PSD95 (green) in presynaptic terminals of photoreceptors. At P7, no abnormal features were observed in the KO retina, and Rs1 is not present. PSD95 was minimally present in the WT or KO retina at P7 (*red arrowheads*), while rhodopsin was present and orderly in both. At P30, the WT retina showed normal maturity, organization of retinal layers, and protein expression of OPN2, Rs1, and PSD95 (*white arrowhead*). The KO retina showed disruption in normal development in the absence of Rs1. Rhodopsin signaling was delocalized, and the PSD95 signal was reduced at the synaptic terminals (*yellow arrowhead*). Scale bar = 40 μm.

**Figure 6 genes-13-01995-f006:**
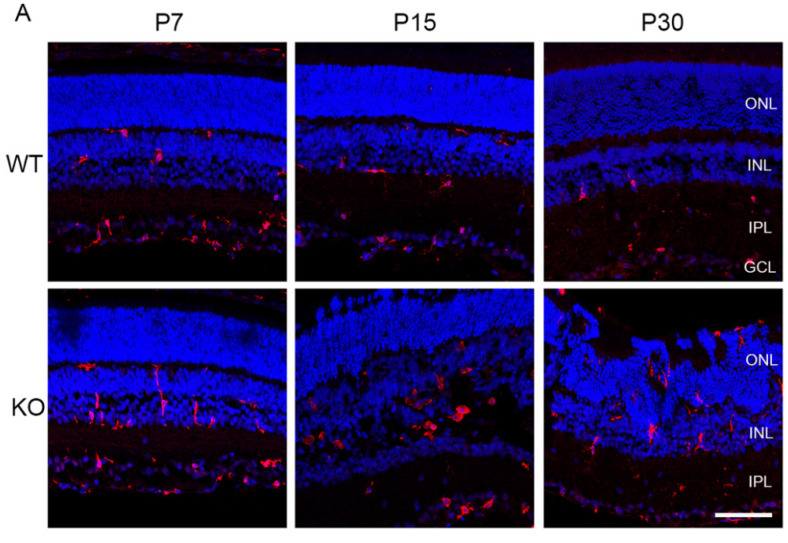
Müller glial activation in *Rs1^-/Y^* rat retina. Retinas of WT and ***Rs1^-/Y^*** rats were immunolabeled for microglia and macrophages (Iba1) and Müller glia (SOX9 and GFAP) at P7, P15, and P30. (**A**) Iba1-IR (red) cells were mainly shown in the ganglion cell layer, inner nuclear layer (GCL, INL) and synaptic layers in the WT retinas. Slightly increased levels of Iba1 cells were found at P7 in the KO retinas. Microglial activation was more noticeable at P15 and P30 of the KO retinas. At P30, Iba1 cells migrated into the ONL and subretinal regions in the KO retinas. Scale bar = 100 μm. (**B**) Upregulated expression of GFAP-IR (green) in Müller glia was evident at P15 and P30 in Rs1-KO. Only the nerve fiber layer (NFL) and GCL in the WT retina showed GFAP labeling. In the KO retina, GFAP expression at P7 was also restricted to the NFL and GCL. Still, between P15 and P30, GFAP labeling gradually extended across all layers, suggesting Müller hypertrophy and reactive gliosis. At P15 and P30, Müller glia nuclei (SOX9, red) were displaced compared to the WT retina. Scale bar = 50 μm.

**Figure 7 genes-13-01995-f007:**
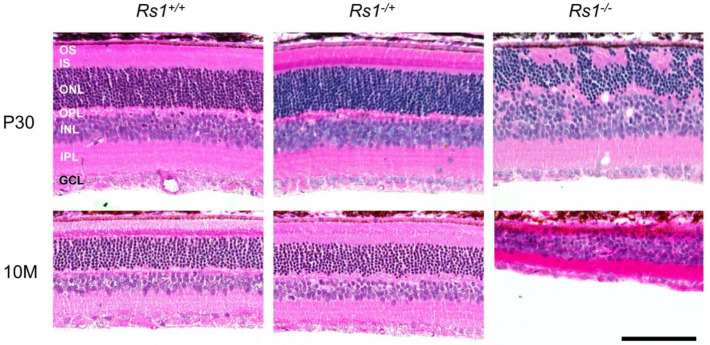
Histology of retinal structural changes of *Rs1h^-/-^* rat in females. Histology of female *Rs1^+/+^*, *Rs1^-/+^* and *Rs1^-/-^* retinas at P30 and 10 months. No structural differences were observed between *Rs1^+/+^* and *Rs1^-/+^* female retinas at the examined time points. Furthermore, at examined time points, no differences were observed at this magnification between male (Figure 2A) and female KOs. The *Rs1^-/-^* showed photoreceptor nuclei dislocated into the photoreceptor inner- and outer- segment (IS/OS) zone, with approx. 40% of the outer nuclear layer (ONL) remaining and a disrupted OPL by P30. At 10 months, the thickness of the OS, IS, ONL, outer plexiform layer (OPL), inner nucelar layer (INL), and inner plexiform layer (IPL) have been reduced; the ONL with only approx. 10% of the ONL remaining. Scale bar = 100 μm.

**Figure 8 genes-13-01995-f008:**
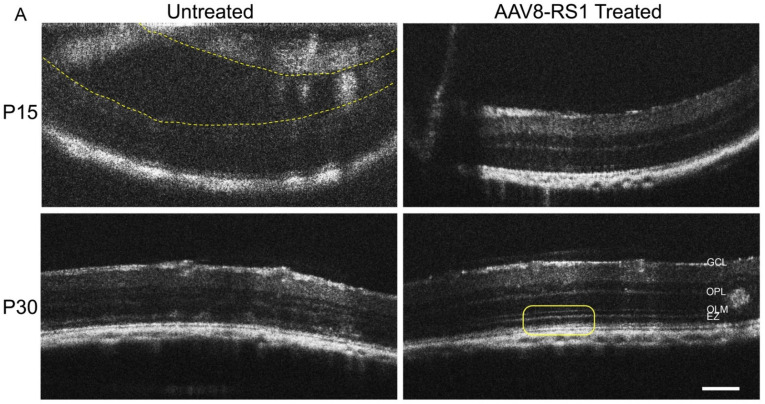
Effects of an intravitreal injection of AAV8-RS1 into the eyes of *Rs1^-/Y^* rat. AAV8 vector expressing RS1, scAAV8-hRS/IRBP-hRS was delivered into the eyes of Rs1-KO rats. Contralateral eyes were untreated and served as the negative control. Intravitreal injection was done at P5 or P6 with 2 x 10^10^ viral genome per eye. (**A**) Optical coherence tomography (OCT) was performed at P15 and P30. At P15, no or substantially reduced formation of retinal cavities was found, and the retinal lamination was well-preserved in the treated eye (N = 7). At P30, eyes that received AAV8-RS1 displayed a distinct laminar structure, including the inner retina, outer plexiform layer (OPL), outer nucelar layer (ONL), and four outer retina reflective bands (ORRBs) (yellow box), while untreated contralateral eyes showed highly disrupted laminar structure. Scale bar = 200 μm. (**B**) Immunohistochemistry for retinal and inflammatory markers was performed at P30. Remarkable rescue effects have been found on the structure of ***Rs1^-/Y^* rat** retinas after AAV8-RS1 gene delivery. Treated retinas showed RS1 expression (purple, *white arrowhead*), with comparable levels to WT retina (Figure 5), on the inner segments of photoreceptors and bipolar cells, while untreated eyes lacked RS1 expression. The treated retina displayed intact OLM (b-catenin, red). Rod outer segments (Rhodopsin, green) were well-preserved. In addition, we demonstrated that RS1 delivery prevented microglial (Iba1, red) and Müller glial (GFAP, green) alterations and cell death of Müller glia (SOX9, red).

**Table 1 genes-13-01995-t001:** Primary Antibodies.

Antibody	Species	Dilution	Product Source
Rs1	guinea pig	1:1000	Custom antibody against N-terminus residues 24–37; Thermo Fisher Scientific, Waltham, MA, USA
Rs1	rabbit	1:1000	Custom antibody against N-terminus residues 24–37; Thermo Fisher Scientific, Waltham, MA, USA
PSD95	goat	1:100	AB12093, Abcam, Cambridge, UK
β-catenin	mouse	1:500	BD 610153; BD biosciences, San Jose, CA
Iba1	rabbit	1:500	019-19741; Wako chemicals, Richmond, VA
OPN2	mouse	1:1000	O4886; Sigma-Aldrich, St. Louis, MO, USA
SOX9	rabbit	1:500	HPA001758; Sigma-Aldrich, St. Louis, MO, USA
GFAP	mouse	1:300	3670S; Cell signaling, Danvers, MA

**Table 2 genes-13-01995-t002:** Difference in onset and progression of exon-1 del Rs1 mouse and rat.

Stage	Phenotype Comparison of Exon-1 Del Rs1 Mouse and Rat
Early (P20)	XLRS mouse retains much better outer retina morphology than XLRS rat. ERG amplitudes are substantially better in XLRS mouse than rat and progressed more slowly.
Mid (3 month)	XLRS mouse retained 70% normal ONL width, whereas rat retained only 3–4 rows of ONL nuclei (30–40% normal thickness). ERG a-wave of XLRS mouse remained near 50% normal, while the a-wave was reduced by 80% at 3 months in XLRS rat.
Late (7 month)	Outer retinal morphology and ONL thickness of XLRS mouse retains more nuclei relative to rat, and the XLRS mouse ERG b-wave response is better preserved than in rat.

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
