# Peer review of "XLRS Rat with Rs1-/Y Exon-1-Del Shows Failure of Early Postnatal Outer Retina Development"

_genes, 2022, doi:10.3390/genes13111995_

Round 1

Reviewer 1 Report

The authors reported on the generation and phenotyping of a rat model for XLRS. The model was evaluation in vivo using OCT and ERG and the tissues were evaluated ex vivo. The authors resported extensive data of this new model and provided insight on the disases’s pathology in rats and compared this with murine models and the human disease. Developing suitable in vivo models is very important for defining a possible therapeutic window and also test possible experimental therapeutics in vivo.

General

I compliment the authors on their work and their presistance in developing a second rat model since they were surprised by the results from their firstly developed rat model. They confirmed their previous results using this different model. Several differences were found between the exon-1 mouse model and the exon-1 (and exon-3) ratmodel. While reading the title and the main text, I was wondering why the authors specifically want to name the mutation Rs1 -/y. They found similar results in homozygous females, so the model also holds true for this group.

Comments, questions and suggestions:

1.       While reading the abstract, many questions popped into my mind. For example: which rat strain was used and which exact mutation was targeted (e.g., deletion of the entire exon1? Or just part of it)? Also: lines 18 and 19 compare this model to the RS1 KO exon-1 mouse model, was it the exact same (homologous) mutation or not? All of these things become clear later but seem important details while reading the abstract.

2.       Lines 14-15: Microglia were active ………. cell activation: What does active mean in this sentence and why is it important? It becomes clear later, but while reading the abstract I was still wondering.

3.       Line 35: encoded by the Rs1 gene: should it be RS1? Since the sentence describes XLRS in humans?

4.       Line 36: Numerous mutations in human RS1 ……. retinal layers. Do the authors refer to the gene with the first RS1 in this sentence? Then place it in italics.

5.       Is the mutation that was introduced based on a specific existing patient population of XLRS? Could the authors clarify a little more on why they chose this exact mutation to recapitulate it in the rat? Or was the reason to compare the mouse model?

6.       Materials and methods section: I miss an alinea about experimental setup with an overview of which experimental groups were used (e.g., genotypes, number per group, which exams, which ages, same animals over time?, sexes etc.).

7.       4.4.: Could the authors explain why differences occurred between the rat and the mouse model? Do they have any clue? Did this happen before in other retinal degeneration rodent models?

8.       Lines 512-514: could there be a genotype/phenotype correlation? In patients, but also in the models?

9.       Lines 523-527: how about the comparison with human patients?

10.   4.6: What did these experiments tell you about the therapeutic window in patients? Will this be similar for all the patients? Is it very early (e.g., should we introduce genetic (prenatally?) testing for this disease to be in time treating it already in the first weeks of life)? How realistic is this possible therapeutic window if in rats it is that early?

11.   Lines 564 – 568: could the points 2 and 3 be combined?

Reviewer 2 Report

-Add to the keywords some spesific words for the research mentioned in the abstract. Now there is few words which actualy are not so spesific just for this research

-Explain in some part of the beginning the meaning of marking style P. Example P7 and P15

-Check all abbreviations: Need to open when mentioned in the abstract as well as in the main text first time e.g. OLM in the line 208

Introduction

”The retinoschisin protein is encoded by the Rs1 gene and plays a key role in XLRS [2]. 35 Numerous utations in human RS1 lead to dysfunctional RS1 protein and disrupt the 36 structural integrity of retinal layers. RS1 

Make all abbreviations similar eg. Rs1 gene vs RS1 and if RS1 means retinoschisin protein add after it (RS1)

- lines 41-48, there is mentioned that there is already many similar mouse models as well as authors origin. It should that`s why justify why still need to publish and produce new model introduced in the present manuscript.

Results

 -  Figure 2 A, mark little bit more clearly ONL, OPL etc into the figure and explain abbreviations in the figure legend   

- Figure 3, make little bit more clearly visible some part of the figure as *

- Figure 6 B, explain also in figure legend B what are red labeled dots, microglia?

- check lines 384-385 and the language of the sentence.

- check line 516

- Is there possible to measure and do as well statistic somehow for histological and immunohistological figures?                                                                                                                                                                                                                                                                                                                  
